# Drimane Sesquiterpene Aldehydes Control *Candida* Yeast Isolated from Candidemia in Chilean Patients

**DOI:** 10.3390/ijms231911753

**Published:** 2022-10-04

**Authors:** Víctor Marín, Bryan Bart, Nicole Cortez, Verónica A. Jiménez, Víctor Silva, Oscar Leyton, Jaime R. Cabrera-Pardo, Bernd Schmidt, Matthias Heydenreich, Viviana Burgos, Cristian Paz

**Affiliations:** 1Laboratory of Natural Products & Drug Discovery, Department of Basic Science, Center CEBIM, Universidad de La Frontera, Av. Francisco Salazar 01145, Temuco 4811230, Chile; 2Departamento de Ciencias Químicas, Facultad de Ciencias Exactas, Universidad Andres Bello, Sede Concepción, Autopista Concepción-Talcahuano 7100, Talcahuano 4260000, Chile; 3Tecnología Médica, Facultad de Salud, Universidad Santo Tomás, Temuco 4780000, Chile; 4Laboratorio de Química Aplicada y Sustentable, Departamento de Química, Facultad de Ciencias, Universidad de Tarapacá, Arica 1000000, Chile; 5Institut für Chemie, Universität Potsdam, Karl-Liebknecht-Str. 24-25, D-14476 Potsdam, Germany; 6Departamento de Ciencias Básicas, Universidad Santo Tomás, Temuco 4780000, Chile

**Keywords:** drimane sesquiterpenoids, *Drimys winteri*, isotadeonal, winterdial, *Candida* yeast, lanosterol 14-α-demethylase, molecular dynamics

## Abstract

*Drimys winteri* J.R. (Winteraceae) produce drimane sesquiterpenoids with activity against *Candida* yeast. In this work, drimenol, polygodial (**1**), isotadeonal (**2**), and a new drimane α,β-unsaturated 1,4-dialdehyde, named winterdial (**4**), were purified from barks of *D. winteri*. The oxidation of drimenol produced the monoaldehyde drimenal (**3**). These four aldehyde sesquiterpenoids were evaluated against six *Candida* species isolated from candidemia patients in Chilean hospitals. Results showed that **1** displays fungistatic activity against all yeasts (3.75 to 15.0 µg/mL), but irritant effects on eyes and skin, whereas its non-pungent epimer **2** has fungistatic and fungicide activities at 1.9 and 15.0 µg/mL, respectively. On the other hand, compounds **3** and **4** were less active. Molecular dynamics simulations suggested that compounds **1**–**4** are capable of binding to the catalytic pocket of lanosterol 14-alpha demethylase with similar binding free energies, thus suggesting a potential mechanism of action through the inhibition of ergosterol synthesis. According to our findings, compound **2** appears as a valuable molecular scaffold to pursue the future development of more potent drugs against candidiasis with fewer side effects than polygodial. These outcomes are significant to broaden the alternatives to treat fungal infections with increasing prevalence worldwide using natural compounds as a primary source for active compounds.

## 1. Introduction

The incidence of serious fungal infections, mainly candiduria and candidemia, has increased significantly during the last three decades, affecting mainly immunocompromised and intensive care unit patients [1,2]. The epidemiology of these candidiases is very dynamic as the prevalence of *Candida* species varies in different geographical areas, kinds of patients and hospital types [3]. One of the major problems of these diseases is the high frequency of non-*albicans Candida* species that present intrinsic or secondary resistance to various antifungals, narrowing medical treatment alternatives and increasing the chances of morbidity and mortality [4]. Today, there is an expected global epidemic of *C. auris* in health care settings with high mortality and multidrug resistance characteristics, with 93% of isolates resistant to fluconzazole, 35% to amphotericin B, 7% to echinocandin, 42% to two or more antifungal classes and 4% to all antifungal classes.

The most important antifungal drugs include amphotericin B, with a polyene structure, and the more recently discovered azole derivatives voriconazole and posaconazole that inhibit ergosterol synthesis interfering with the fung2al cell membrane [5], and echinocandins as caspofungin, anidulafungin and micafungin that inhibit β-1,3-d-glucan synthase interfering with fungal cell wall synthesis [5]. The most known mechanisms for antifungal drug resistance are *ERG3* gene mutation and reduction of ergosterol for polyenes, efflux by multidrug transporters for azoles, *ERG11* gene mutation, decrease in affinity, alteration in the ergosterol biosynthetic pathway, and mutation in the *FKS1* and *FKS2* bindings units for echinocandins [6,7,8]. There are still major weaknesses in the spectra, potency, safety, and pharmacokinetic properties of these agents. In addition, the emergence of fungal strains resistant to existing antifungal drugs, especially fluconazole, is becoming a significant problem for human health worldwide.

*Drimys winteri* J.R Forst. and G. Forst. is a tree member of the Winteraceae family. Commonly called Canelo, this tree is considered sacred for the *Mapuche* people in the center-south of Chile, and the *Machi* or Mapuche healer use its leaves and barks for many disease treatments. *Drimys winteri* possesses several biological properties due to drimane sesquiterpenoids produced in its leaves and barks, including trypanocidal activity [9], insecticidal activity [10], inhibition of bacterial Quorum Sensing [11,12] and inhibition of the α4β2 Nicotinic Acetylcholine Receptors [13] involved in the drug addiction and depression states. *D. winteri* extracts are strongly pungent, irritating for eyes, sensitive skin or injuries due to the presence of polygodial, an α,β-unsaturated dialdehyde with agonist activity of the receptor hTRPA1 [14] and the potassium channels TASK-1 (K_2P_ 3.1), TASK-3 (K_2P_ 9.1), and TRESK (K_2P_ 18.1), which are targets for pungency sensations [15]. Polygodial also inhibits voltage-gated sodium channels Nav1.7 and Nav1.8 involved in pain sensation with an IC50 of 16 ± 8 and 57 ± 7 µM, respectively [16]. Moreover, polygodial displays potent antifungal activities against a broad spectrum of filamentous fungi including *Botrytis cinerea* [17], *Gaeumannomyces graminis* var. tritici [18] and parasitic yeast as *Candida albicans* with an IC50 of 3.13 µg/mL [19]. This activity is also enhanced in mixtures with anethole or miconazole. Its activity has been explained by the capability to act as a nonionic surfactant that disrupts the lipid–protein interface [20,21,22,23]. Further developments in this subject can be expected from structurally related compounds extracted from *D. winteri* that have not been characterized so far and offer valuable opportunities to address the challenges of candidiases treatment. In this regard, we previously explored the allylic oxidation of isodrimeninol with pyridinium chlorochromate (PCC), producing five drimane derivatives with inhibitory activity of the enzyme lanosterol 14-alpha demethylase, which plays a key role in the production of ergosterol in fungi cells [24]. In this report, we present the isolation of four natural drimane sesquiterpenoids from *D. winteri* barks including polygodial, isotadeonal, an epimer of polygodial but without pungent sensation, a new α,β-unsaturated 1,4-dialdehyde, which we call winterdial and the sesquiterpene alcohol drimenol, which was the starting material for the synthesis of drimenal. These compounds were evaluated in vitro against six Candida strains—*C. albicans*, *C. krusei*, *C. glabrata*, *C. topicalis*, *C. parapsilosi* and *C. lusitaneae*—which were isolated from candidemia patients of Chilean hospitals and by in silico analysis in the enzyme lanosterol 14-alpha demethylase.

## 2. Results

### 2.1. Identification of Yeast Samples by MALDI-TOF-MS

The yeast identification was corroborated previously by morphological analysis and further protein fingerprint by MALDI-TOF-MS of fresh colonies. Previously to the analysis, the equipment was calibrated with the protein calibration standard I (insulin, ubiquitin, cytochrome C and myoglobin). The protein fingerprint of each yeast was compared to a library of 1301 spectra of fungal identifications in the range of *m/z* 3000–15,000. The results of identifications are presented in the Table 1 as logarithmic scores between 0 and 3.0. Scores above 1.7 were automatically higher for identification at the gender level, and above 2.0 for species.

### 2.2. Secondary Metabolites Isolated from Drimys winteri

The purification of the total extract of *D. winteri* gave five drimane sesquiterpenoids identified as: drimenol (0.04% yield), isotadeonal (**2**, 0.0624% yield), polygodial (**1**, 0.092% yield), and a new compound that was named winterdial (**4**, 0.0012% yield) (Figure 1). The structure of these compounds was determined by TLC of pure standards and ^1^H- and ^13^C-NMR characterization, as detailed in Table 2 and Table 3 [25]. Winterdial (**4)** is a new α,β-unsaturated sesquiterpene dialdehyde, corresponding to (*E*)-(1*R*,4a*R*,8*R*,8a*S*)-7,8-diformyl-4a-hydroxy-4,4,8a-trimethyl-1,2,3,4,4a,5,8,8a-octahydronaphthalen-1-yl-3-(4-hydroxyphenyl)acrylate. This compound showed a specific rotation of [α]_D_^20^ + 52.3547 (*c* 0.2650, acetone) and an HRESIMS (M + 1) signal of 413.1981, corresponding to the molecular formula C_24_H_28_O_6_.

### 2.3. Oxidation of Drimenol with Pyridinium Chlorochromate

The fraction F2 (*n*-hexane/EtOAC 9:1 *v*/*v*) produced the drimane sesquiterpenoid alcohol drimenol by cryocrystallization at 4 °C. Oxidation of drimenol with 1 equivalent of pyridinium chlorochromate (PCC) in CH_2_Cl_2_, at 0 °C, in short times, produced drimenal (**3**) with an 85% yield [26]. Longer reaction times, at room temperature, produced the 11-nor-drimane derivative **5** as was reported by Cuellar et al. [27]. The structures of compounds **3** and **5** are given in Figure 1. Compound **3** was characterized by ^1^H-NMR and ^13^C-NMR as summarized in Table 2 and Table 3.

### 2.4. Anti-Candida Activity

Quantitative activity data for compounds **1** to **4** against the Candida strains *C. albicans*, *C. krusei*, *C. glabrata*, *C. tropicalis*, *C. parapsilosis* and *C. lusitaneae* were obtained by the broth microdilution method for the determination of minimum inhibitory concentration (MIC) values. Additionally, minimum fungicidal concentrations (MFC) values were obtained by colony formation from the MIC´s wells seeded on a Petri dish with Müller–Hilton agar. Stock solutions (1 mg/mL) were prepared by dissolving the tested compounds and the reference antifungal drug, fluconazole, in sterile DMSO. The obtained values are presented in Table 4. Analyses were repeated three times and in triplicate. 

### 2.5. In Silico Studies

Blind molecular docking calculations were performed using the CB-dock server to evaluate whether the binding of the ligands to the catalytic site of lanosterol 14-alpha demethylase is predicted as the top-ranked pose after the curvature-based cavity detection and ligand docking protocols implemented in the server. In all cases, the ligands were predicted to bind to the catalytic pocket with the highest score within the set of sampled conformations (Figure 2A). The dynamic behavior of the docked structures was then examined through 150 ns unrestrained molecular dynamics simulations at 300 K. Systems were previously minimized, heated and equilibrated following a standard simulation protocol, after which equilibration was verified through the stabilization of the potential energy, kinetic energy, volume and temperature of each system. Trajectory analysis was performed on 300 frames retrieved from the 150 ns production dynamics. Results revealed that all ligands remained bonded in the deepest region of the catalytic pocket nearby the HEME group. Root-mean-square deviations (RMSD) for the ligands in protein-aligned trajectories show narrow distributions for all systems with median values between 1 and 3 Å, thus indicating very low mobilities for the compounds under study (Figure 2B).

Intermolecular contact analysis was carried out to identify the protein residues that remain in close contact with each ligand at distances < 3.0 Å throughout the simulated trajectories. Results confirm that all ligands occupy a similar region of the binding pocket and that their recognition is mostly mediated by non-polar and aromatic residues of the cavity such as Phe129, Tyr135, Phe231, Gly309 and Leu375, Figure 3.

Binding free energy calculations under the MM/GBSA approach were applied to estimate the strength of the ligand–protein association. All ligands resulted in binding free energies ranging between −30 and −36 kcal/mol without significant variations among the set of complexes, which indicates a similar capacity to bind to the enzyme, Figure 4A. Energy decomposition was applied to obtain the gas-phase components for the binding free energy in the set of complexes under study, Figure 4B. Gas-phase binding free energies revealed that the ligands’ interactions with the binding pocket are mostly mediated by van der Waals contacts, with a minor role of electrostatic terms. These findings suggest that hydrophobicity could play a crucial role to enable an effective inhibition of the target enzyme. To address this aspect, ligand desolvation free energies were calculated for compounds **1**–**4,** resulting in a substantial increase for winterdial (**4**) compared with compounds **1**–**3**, Figure 4C. 

## 3. Discussion 

In recent years, drimane sesquiterpene aldehydes have shown potent activities in the control of *Candida* yeast. Particularly, polygodial succeeded in controlling *Candida albicans* and non-*albicans* yeasts with IC50 values of approximately 3 µg/mL [19]. Moreover, this compound acted as an enhancer of fluconazole or anethole activity. Despite its potent properties, polygodial is irritant to sensitive skin such as tongue or eyes, which makes it unsuitable for therapeutical use. In the search for new drimane sesquiterpenoids to control *Candida* yeast, herein we reported the isolation and characterization of four natural drimane sesquiterpenoids including drimenol, polygodial (**1**), isotadeonal (**2**) and a new compound named winterdial (**4**). Isotadeonal (**2**) is an epimer of polygodial with better activities against non-*albicans* species such as *C. kursei*, *C. glabrata*, *C. tropicalis and C. lusitaneae*, and with results comparable to fluconazole in the control of *C. glabrata*. Moreover, this compound is 4-fold more active for the control of *C. krusei* than fluconazole, which is a valuable outcome to continue exploring isotadeonal as a molecular scaffold to develop novel compounds with activity against candidiases. All drimane sesquiterpene aldehydes showed fungostatic activity and only compounds **2** and **3** displayed fungicidal activity at concentrations approximately 25 to 60 µg/mL. The new sesquiterpene dialdehyde **4** is the less active compound of the group. This observation calls into question the hypothesis that explains the promiscuous activity of polygodial (**1**) against several receptors just by a high reactivity against nucleophile residues given by the α,β-unsaturated 1,4-dialdehyde moiety, such as lysine, forming pyrrole derivatives [28] or with cysteine involving thiol-Michael adduct formation by bimolecular nucleophilic substitution [29,30,31]. In these cases, compound **1** could produce a covalent inhibition in multiple proteins, and the lipophilicity of the structure could act as a nonionic surfactant that disrupts the lipid–protein interface [20,21,22,23]. Compound **4** is closely related to **1** in the second ring of the structure, and shares the same capability of forming pyrrole or sulfur derivatives. Nevertheless, it exhibited very low activities against all Candida evaluated. 

Molecular modeling studies were carried out to examine the potential of compounds **1** to **4** to bind to the active site of the enzyme lanosterol 14-alpha demethylase, which is a proposed target for their antifungal activity. According to our findings, all compounds are capable of binding to the active site of the enzyme with similar binding free energies, which support a potential molecular mechanism of action through the inhibition of ergosterol synthesis. In this regard, the lower in vitro activity of compound **4** could be explained by the large energetic cost for ligand desolvation to enable its entrance to the buried catalytic pocket of the enzyme, which could be detrimental for achieving the expected inhibitory mechanism. This high energetic cost arises from the ester and hydroxyl moieties of compound **4**, which provide polar sites for favorable intermolecular contacts with the solvent.

The antifungal activity of drimane sesquiterpenoids could go beyond fungal wall disruption or inhibition of ergosterol synthesis. For example, they can affect biofilm formation and dimorphism in *Candida* yeast as reported by Xie et al. in 2015 [32]. This aspect goes beyond the scope of this work and will be evaluated as part of our future attempts in this field.

The results herein presented highlight the potential of natural and semi-synthetic drimane sesquiterpene aldehydes as valuable molecular scaffolds for the development of novel anti-candida compounds. Particularly, isotadeonal (**2**) appears as a valuable alternative for replacing polygodial as a lead compound in the search for novel antifungal drugs, as it exhibits higher in vitro activity with fewer side effects arising only from the epimerization at C9 of compound **1**. We believe that the results herein presented will be valuable to future attempts to develop alternative therapies for fungal diseases with increasing prevalence worldwide.

## 4. Materials and Methods

### 4.1. Pathogens

Candida strains were obtained from the Culture Collection of the Mayor University at Santiago, Chile [33]. Each strain was transferred from a cryo-tube to a Petri dish with Sabouraud-dextrose agar and incubated at 35 ± 2 °C for 48 h. Two or three colonies were transferred to a chromogenic media and incubated at the same conditions to verify purity and species by proteome analysis.

### 4.2. Proteome Fingerprinting of Candida Yeast

Samples of yeast were incubated in the dark at 37 °C for 48 h, then colony samples were applied directly to a sample plate of the equipment, following the procedures as described by Gobom et al., 2011 [34], where the sample was coated with a saturated solution of α-cyano acid 4-hydroxy cinnamic diluted in 50% ACN with 2.5% TFA. The mass spectra were used using a MALDI-TOF-MS Autoflex Speed (Bruker Daltonics, Bremen, Germany) equipped with an intelligent beam laser source (334 nm). Analysis was carried out in the linear mode with positive polarity, an acceleration voltage of 20 kV and a delay extraction of 220 ns. Each spectrum was collected as an average of 1200 laser shots with enough energy to produce good spectra without saturation in the range of *m/z* from 2000 to 20,000. Before analysis, the equipment was externally calibrated with the protein calibration standard I (Bruker Daltonics, Bremen, Germany; insulin, ubiquitin, cytochrome C and myoglobin) with FlexControl 1.4 software (Bruker Daltonics, Bremen, Germany). Analysis was carried out with the MALDI Biotyper Compass 4.1 software (Bruker Daltonics, Bremen, Germany) in the range of *m/z* 3000–15,000 compared to a library of 1301 spectra of fungal identifications.

### 4.3. Vegetal Material

Barks of *D. winteri* were collected in Temuco, IX Region of Chile, in December 2020. An amount of 1 kg of bark tree was dried in an oven at 50 °C for 2 days, then the material was milled in a size less than 2 mm, and the powder material was extracted with ethyl acetate (EtOAc, 2 L) for 3 d, at room temperature. The organic layer was filtered and evaporated in vacuo giving a total extract 450 g (after 2 extractions).

### 4.4. General Information

Analytical thin-layer chromatography (TLC) was carried out on Merck Silica Gel 60F254 sheets (Darmstadt, Germany). It was employed for monitoring the purification process and the reaction progress of new compounds. The elution was using a mixture of n-hexane: EtOAc = 4:1 and UV light (254 nm) together with molybdophosphoric acid and heating for visualization. Preparative chromatography was performed using Merck silica gel 60 and Sephadex LH-20 (25–100 μm; Aldrich, Santiago, Chile). Solvents and fractions were concentrated in a Büchi R100 rotavap. Solvents used in this study were distilled before use and dried over appropriate drying agents.

### 4.5. Purification of Drimane Sesquiterpenoids

The total organic extract was fractioned by flash column chromatography (CC) using a mixture of solvents from hexane to EtOAc, giving 8 fractions (F1–F8). From the F2 a white solid was separated after solvent concentration, identified as drimenol (400 mg, crystals, 0.04% yield). Looking for compounds that contain the aldehyde moiety, an exploratory NMR was measured for each fraction, giving that F3, F4 and F7 have ^1^H-signal between 9 and 10 ppm, attributable to aldehydes. Further purification of F3 with hexane/EtOAc (4:1 *v*/*v*) gave isotadeonal (624 mg, yellowish oil, 0.0624% yield). From F4 was purified polygodial (920 mg, yellow oil, 0.092% yield), and the purification of F7 with hexane/EtOAc (1:4 *v*/*v*) gave winterdial (12 mg, white powder, 0.0012% yield). The compounds were confirmed by NMR, using pure standards previously characterized by spectroscopic methods. 

### 4.6. Oxidation of Drimenol with Pyridinium Chlorochromate

A solution of drimenol (222 g/mol, 100 mg, 0.45 mmol) in dichloromethane (10 mL) was cooled at 0 °C in an ice bath and mixed with 1 equivalent of pyridinium chlorochromate (PCC, 215.5 g/mol, 97 mg) added dropwise. Then, the reaction mixture was stirred at 0 °C for 60 min, till all the starting material disappeared by TLC monitored each 15 min. The solvent was removed under vacuum and the gummy residue was further purified by CC with hexane/EtOAc (9:1 *v*/*v*). The corresponding drimenal was purified as colorless oil in the less polar fraction (Rf: 0.8 hexane/EtOAc 9:1 *v*/*v*, 220 g/mol, 84 mg, 85% yield).

### 4.7. NMR Analysis of Drimane Sesquiterpenoids

The structures of compounds **1** to **4** were elucidated by 1D and 2D NMR. The ^1^H- and ^13^C-NMR spectra were recorded in CDCl_3_ solution in 5 mm tubes at RT on a Bruker Avance III 600 MHz spectrometer (Bruker Biospin GmbH, Rheinstetten, Germany) at 600.13 (^1^H) and 150.61 (^13^C) MHz, with the deuterium signal of the solvent as the lock and TMS (for ^1^H) or the solvent (for ^13^C) as internal standard. All spectra (^1^H, ^13^C, gs-H, H–COSY, edited HSQC, and gs-HMBC) were acquired and processed with the standard Bruker software (TopSpin 4.x, Bruker, Germany).

### 4.8. Antifungal Assay against Candida Species

Candida strains *C. tropicalis*, *C. parapsilopsis*, *C. krusei*, *C. lusitaneae*, *C. albicans* and *C. glabrata* were acquired from the University Mayor in Chile. The in vitro antifungal assay was performed in triplicate by a broth microdilution method following Clinical Laboratory Standard Institute (CLSI) recommendations according to the document M27A. In summary, the samples were dissolved in dimethyl sulfoxide (DMSO) stock solution at 1200 μg/mL and then diluted with sterile water to a final concentration of 120 μg/mL. In 96-well microplates, this solution achieves concentrations between 60.00 and 6.25 μg/mL by serial dilutions in culture media (RPMI) to 100 µL. The inoculum (100 µL) was adjusted to yield a cell concentration of 2.5–5.0 × 10^5^ UFC/mL, and the final DMSO content was 5% (*v*/*v*). One inoculated well was included as microorganism growth control. One non-inoculated well was included to ensure medium sterility. The plates were incubated at 37 °C for 48 h in a humid chamber. Then, the growth in the positive control column and its absence in the negative control were visually verified. The MIC values were determined at 24 h, as the lowest concentrations of each compound capable of inhibiting microorganism growth by visual inspection, compared to growth control. The MFC or minimum fungicidal concentration was determined by seeding 20 µL of each well from which the MIC assays were done, on a Petri dish with Müller–Hilton agar. The Petri dish was incubated at 37 °C for 48. The MFC corresponds to the lowest concentration capable to inhibit colony formation by visual inspection after MIC assay. MIC and MFC were determined by triplicate and each assay was done three times using different starting yeast. Fluconazole was used as a positive control at a concentration of 50, 25, 12.5, 6.25, 3.13, 1.56 and 0.78 µg/mL.

### 4.9. Molecular Simulation

The initial coordinates of the enzyme and its natural substrate (lanosterol) were retrieved from the PDB database under code 4LXJ [35]. The protein structure was checked to fix protein discontinuities considering the UNIPROT code P10614 and protonation states were set to pH 6.5 using the H++ web server [36]. The initial coordinates of polygodial, isotadeonal, winterdial and drimenal were built using the Avogadro software and minimized using the MMF94 force field with 3500 steepest descent steps and a convergence criterion of 10^−7^ [37]. Blind ligand–protein docking was conducted for each compound against the structure of lanosterol 14-alpha demethylase using the CB-Dock web server [38,39]. The highest-ranked binding mode for all ligands corresponds to the association with the lanosterol cavity in the proximity of the HEME group. The resulting coordinates were used as input structures for molecular dynamics simulations using the AMBER20 software [40]. To this aim, force field parameters consistent with the GAFF force field were obtained for each ligand using AM1-BCC charges in Antechamber. HEME parameters were retrieved on May 2022 from the AMBER parameter database available at http://amber.manchester.ac.uk/. The protein was modeled using the ff19SB forcefield. Simulated systems were solvated in an octaedric box of OPC waters extended 10 Å from the outermost atom of the complex. Systems were neutralized with proper counterions to keep charge neutrality. MD simulations were performed using the pmemd.CUDA program using the following protocol: (a) 1500 steepest descent minimization steps followed by 3500 conjugate gradient minimization steps for water molecules relaxation, (b) 1500 steepest descent minimization steps followed by 6500 conjugate gradient minimization steps for the entire system, (c) 500 ps of progressive NVT heating from 0 to 300 K, (d) 500 ps of NVT equilibrium at 300 K with restrains applied to the protein backbone, (e) 20 ns of NVT equilibrium at 300 K, and finally (f) 150 ns of unrestrained NPT production dynamics at 300 K and 1 bar from which production data were collected. During MD simulations the cutoff for non-bonded terms was 10 Å, long-range electrostatics were treated using the Particle-Mesh Ewald approach, and the SHAKE algorithm was employed to constrain all bonds involving hydrogen. Trajectory analysis was carried out using CPPTRAJ software. MM/GBSA binding free energy calculations were carried out to estimate the strength of protein–ligand interactions using the MMPBSA.py module in Amber20, under a single trajectory approach [41]. GB calculations were carried out using the modified GB model (igb = 5) with mbondi2, and α, β, and γ values of 1.0, 0.8, and 4.85, respectively. Dielectric constants for the solvent and the protein were set to 80 and 1, respectively. A salt concentration of 0.15 mol L^−1^ was considered in this process to mimic physiological conditions for binding free energy estimates. The entropic term was not included in our calculations due to the size of the systems under study.

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
