# Peer review of "Drimane Sesquiterpene Aldehydes Control *Candida* Yeast Isolated from Candidemia in Chilean Patients"

_ijms, 2022, doi:10.3390/ijms231911753_

Round 1
Reviewer 1 Report
Three known and one new drimane type sesquiterpenoids were isolated and tested for antifungal activity. Topic and results are interesting, following the general procedures of of natural products isolation and bioassays. The products are well characterized. The manuscript is well written. It meets the standards. It is accpted for publication in IJMS as is.
Author Response
I would like to thank to the reviewer for his/her time and accept the MS as is.
Dr. Cristian Paz
Reviewer 2 Report
General Impression
The authors present a study on the anti-Candida sp. effects of fractionated organic extracts of Drymis winteri. The authors focus on four terpenoid compounds, which they characterize with spectroscopic methods, and establish their fungistatic and fungicidal properties on several clinically relevant Candida species. The authors also include a molecular modelling study to explain the interactions of the extracted compounds with the supposed target, lanosterol 14-alpha demethylase. The manuscript is well written, the experiments are conducted thoroughly, methods are described appropriately, and the data interpretation is justified by the results. The authors conclude that of the examined compounds, isotadeonal has the highest activity against the six tested Candida sp., but that the different inhibition profiles of structurally similar compounds suggest that the mechanism of action of these molecules might be more complex than previously assumed.
Suggestions for improvement
The study has no shortcomings in the area experimental procedures and data interpretation but could benefit from the inclusion of additional data on the action of their compounds. Sesquiterpenes in plant extracts influence Candida sp. biology much beyond a simple inhibition of ergosterol synthesis, affecting the fungus’ morphological transition and biofilm formation (see e.g. Xie et al., 2015, Sesquiterpenes from Carpesium macrocephalum Inhibit Candida albicans Biofilm Formation and Dimorphism, Bioorganic & Medicinal Chemistry Letters 25(22)). The influence of the compounds isolated in the present study on morphology and biofilm formation has not been examined, and the task might be beyond the scope of the manuscript. However, the paper could benefit from adding this aspect to the discussion and to future directions.
Author Response
Response. We appreciate this valuable comment and the favorable opinion of the Reviewer on our work. As per her/his suggestion, we have included a new statement in the Discussion section, pointing out the aspects indicated by the Reviewer:
“The antifungal activity of drimane sesquiterpenoids could go beyond fungal wall disruption or inhibition of ergosterol synthesis. For example, drimane sesquiterpenoids can affect biofilm formation and dimorphism in Candida yeast (Xie et al., 2015). This aspect goes beyond the scope of this report and will be evaluated as part of our future developments in this field.”

Reviewer 3 Report
Drimane sesquiterpene aldehydes control Candida yeast iso- 2 lated from candidemia in Chilean patients
In this manuscript Marin et al. purified drimenol, polygodial , isotadeonal, and a new drimane α,β-un-saturated 1,4-dialdehyde (winterdial) from barks of D. winteri. Moreover, the oxidation of drimenol produced the monoaldehyde drimenal. These 4 aldehyde sesquiterpenoids were evaluated against 6 Candida species isolated from candidemia patients in Chilean hospitals. Results showed that 2 displays fungistatic activity against all yeasts and its non-pungent epimer. Molecular simulations suggest that compounds appear to be capable of binding to the catalytic pocket mostly through hydrophobic contacts with similar binding free energies. This study is of interest, my suggestion is minor revision.
First, the figures and tables can be improved and revised to be more informative, especially regarding layout and presentation. The legends should be improved for clarity and comprehensible. Authors findings can help in future microbial works. In this regard, provide a comprehensive conclusion and future perspectives, including main text and abstract.
Second, the entire manuscript has paragraphs that do not have enough citations for information the author has written. A careful review should be carried out.
Lines 46-47 = Please, add references
Lines 48-49 = Please, add references
Lines 60-62 = Please, add references
Line 66= Please, add references (Inhibition of the α4β2 Nicotinic Acetylcholine)
Line 77: 3.13 ug/mL (Lee et al., year?)
Line 83: PCC?
Line 99: “chromogenic media and incubated at the same conditions to verify species and purity”. Please, describe more details about this morphological analysis
Lines 332-336 Please, add references
Author Response
The authors sincerely thank the reviewers for their valuable comments on our work. In the revised version of the manuscript, we carefully addressed each comment and provided a detailed response in this document. Additionally, careful English language and grammar revisions were conducted to correct the minor mistakes pointed by the referees after the revision process.
- First, the figures and tables can be improved and revised to be more informative, especially regarding layout and presentation.
Response. Figures and Tables were modified to improve the clarity and informative purpose. Titles and captions were also corrected to provide better descriptions and straightforward information to the readers. Authors sincerely thank Reviewer 2 for pointing out this aspect.
- The legends should be improved for clarity and comprehensible.
Response. Figure legends and Table titles were modified as per the Reviewer’s suggestion. We sincerely hope that the current versions provide clear and comprehensible information for the readers.
- The authors findings can help in future microbial works. In this regard, provide a comprehensive conclusion and future perspectives, including the main text and abstract.
Response. The manuscript has been carefully revised to include more comprehensive conclusions and future perspectives, including the main text, and abstract. In this regard, we have highlighted the relevance of compound 2 as a non-pungent drimane sesquiterpenoid with valuable properties as a molecular scaffold for the future development of drugs to treat fungal infections with increasing prevalence worldwide
- Second, the entire manuscript has paragraphs that do not have enough citations for information the author has written. A careful review should be carried out.
Response. The entire manuscript has been revised and the following references were included as per the reviewer’s suggestions:
Lines 46-47. Krcmery, V., & Kalavsky, E. (2007). Antifungal drug discovery, six new molecules patented after 10 years of feast: why do we need new patented drugs apart from new strategies?. Recent patents on anti-infective drug discovery, 2(3), 182-187.
Lines 48-49. Krcmery, V., & Kalavsky, E. (2007). Antifungal drug discovery, six new molecules patented after 10 years of feast: why do we need new patented drugs apart from new strategies?. Recent patents on anti-infective drug discovery, 2(3), 182-187.
Lines 60-62. We added references at the end of the corresponding paragraph: (Krishnasamy et al., 2018; Urbanek et al., 2022; Al-Baqsami et al., 2020)
Line 66. The reference of (Arias et al., 2018) was added
Line 77: 3.13 ug/mL (Lee et al., 1999). The year corresponding to the reference was corrected in the revised version of the manuscript.
Line 83: The acronym PCC was changed to pyridinium chlorochromate (PCC)
Lines 332-336. We added the reference (Lee et al., 1999)
- Line 99: “chromogenic media and incubated at the same conditions to verify species and purity”. Please, describe more details about this morphological analysis
Response. The purity of yeast was evaluated by microscopy, just trying to avoid bacteria contamination, but the species were evaluated by MALDI TOF analysis, which is explained in the 2.2. Proteome Fingerprinting of Candida yeast
